# The Impact of the Minimum Wage on Employment: An EU Panel Data Analysis

**Cristian Valeriu Paun [1], Radu Nechita [2], Alexandru Patruti [2,* and Mihai Vladimir Topan [2,***

[1] Department of International Business and Economics, Bucharest University of Economic Studies, 010374 Bucharest, Romania; cristian.paun@rei.ase.ro

[2] Department of European Studies and Governance, Babes-Bolyai University, 400090 Cluj Napoca, Romania; radunechita@gmail.com

\* Correspondence: alexandru.patruti@rei.ase.ro (A.P.); topan_vlad@yahoo.com (M.V.T.)

**Abstract:** Minimum wage laws have become one of the most debated state interventions in the economy, being considered by many specialists as a very efficient tool used to correct certain labour market failures. The aim of this paper is to explore the relationship between minimum wage and employment dynamics, with a special focus on some vulnerable categories recognized in the literature (young people, female workers, the elderly, etc.). Thus, we analysed the relation between the dynamics of minimum wages and that of employment in 22 EU countries, panel data (1999–2016). The results suggest a negative impact of the minimum wage on total employment and on sensitive categories (youth, female workers, the elderly). The long-running negative impact holds for all but one group (55–64 years). The models were tested for random and fixed effects and the results were correspondingly adjusted with country and time and random and fixed effects. Cointegration tests and the tests using lagged minimum wage also confirm a robust relationship between the dynamics of the minimum wage and that of employment over time. Our findings are consistent with many previous studies and confirm the recommendations to prudently use this public policy tool.

**Keywords:** minimum wage; employment; labour market

## 1. Introduction

According to Walter Heller, "an economist is a man who, when he finds something that works in practice, wonders if it works in theory." This half-joke about economists is arguably true for many topics, but does not hold anymore in the field of minimum wage. Indeed, there is a growing number of papers for which "what does not work in theory could work in practice". Somewhat of a consensus among economists—as reflected by the most popular textbooks—was that the imposing of a minimum wage above the labour market price would have the same consequence as any other (effective) price floor on any other market. By employing a simple supply and demand model, one can quickly show that imposing a minimum wage above the equilibrium level will decrease the demand for labour while increasing supply. Involuntary unemployment is cited as one of the most important consequences of minimum wages.

However, this quasi-consensus is not reflected in public policies: most countries are enforcing some forms of minimum wage. Moreover, during the last couple of decades, this "economists' quasi-consensus" was challenged inside the profession on empirical grounds. Our research aims to identify some consequences of minimum wage regulation on employment, especially on sections of the population most likely to be adversely affected by its enforcement. The five research hypotheses proposed (see Section 3.1 below) are subsumed to our research objectives: to study the general impact of minimum wage dynamics on the dynamics of general employment; to identify the impact on selected

vulnerable categories of employed persons such as the young, female workers, the elderly; and to test the long-term relation between the dynamics of minimum wage and that of employment, including those selected vulnerable categories. The paper is organized as follows: the literature review (1) introduces our research (2) and its empirical results (3); policy implications and recommendations conclude the paper (4).

## 2. Literature Review

According to one of the most popular economics textbooks of all times: "[a]lthough almost everyone would agree that a living wage requires higher pay, studies show that a high minimum wage often hurts those it is designed to help […] The minimum wage probably raises the income of some low-wage workers at the expense of others who cannot find work or consumers who must pay higher prices'' [1]. Since the mid-1990s, a wave of empirical studies concluded that (moderate) increases in minimum wage have no significant effect on employment. This had consequences for political debates. It was reflected in mass media and forced the economic profession to a rather fast reaction. On the other hand, there were also studies of minimum wage increases which identified effects in line with standard economic theory. The debate on the topic is reflected in economic textbooks, despite the acknowledged difficulties of their revision [2]. For example, in the latest edition of Samuelson and Nordhaus' *Economics*, minimum wage is mentioned 55 times. It is also significant that the topic is treated in deeper detail under the title "The minimum wage controversy". Mankiw [3] reassures the students that this controversy has not shaken the whole profession: 79% of economists agree with the idea that "a minimum wage increases unemployment among young and unskilled workers'', placed seventeenth among the economic policy propositions most consented among economists. In the comprehensive *Handbook of Labour Economics*, Brown [4] reviewed the literature generated by the renewed interest in minimum wage and remarked that "…recent work suggests that a relative consensus on the effects of the minimum wage on employment came undone". Some authors, such as Brecher and Gross [5], have identified (theoretical) conditions under which "within a simple general-equilibrium model of perfect competition, higher minimum wages may paradoxically lead to greater levels of total employment". They introduced heterogeneous households with specific propensities to consume, using a two goods model. One is relatively capital intensive while the other is labour intensive. Their claim was that the negative side effects of supply can possibly be compensated by a positive side effect in demand, caused by the specific propensities of the households to consume: "income-redistribution effects larger than substitution effects in production and consumption." However, the study appears to be no more than a curiosity, a "theoretically interesting possibility", as the authors themselves claim (p. 169). The level of specificity of such a case is most likely too narrow to approximate real life conditions. A much more interesting question regarding this issue would be not whether minimum wages cause dis-employment effects, but whether they cause an increase in workers' income in the long run. Economides and Moutos [6] argued that this is not the case. They used an intertemporal model of capital accumulation with two agents (workers and capitalists) to show that a minimum wage acts like a tax on capital which will reduce the future aggregate income received by workers. There are some specific conditions under which the workers who remain employed after the introduction of the minimum wage do have a higher income, but this can only come at the expense of the ones left unemployed. Moreover, this situation cannot be solved with the aid of redistribution, as even if the latter would receive unemployment pay-offs, these would be lower than the free market wage. Nevertheless, the lack of consensus in this specific area of economics is not caused by theoretical arguments, but by the diversity of the findings of empirical research. As interesting and important as they are for economists, theoretical considerations seem to have lower relevance than empirical studies in shaping public opinion and, indirectly, political decisions. With the passage of time, more sophisticated methods of measuring the effects of minimum wages have evolved, and

many claim that a "new consensus" has been reached [7]. Card and Krueger's [8] is probably the most notorious paper which does not agree with the idea that minimum wages negatively affects employment, even in the case of young adults. Their research in the study consisted of the food and beverage industry in New Jersey and showed that an increase in the minimum wage level actually increased employment. In a later book, they presented evidence that an increase in the federal minimum wage level in the US had no significant effect on the level of employment (a less "paradoxical" conclusion). However, their research is highly controversial and it is doubtful that their case study from New Jersey, even if methodologically correct, can be generalized at national level (see also the comments of Bazen [9]. Neumark and Wascher [10] (p. 123) have provided a comprehensive review of this "new minimum wage literature" and conclude that the research available at that time "…when read broadly and critically [is] largely solidifying the conventional view that minimum wages reduce employment among low-skilled workers, and as suggesting that the low-wage labour market can be reasonably approximated by the neoclassical competitive model."

New statistical and econometric methods do not necessarily generate a consensus on the issue of minimum wages. For example, a recent article by Wang et al. [11] that employed a C-lasso technique in order to estimate the impact of the minimum wage on the employment rate managed to conclude that "the findings also reveal substantial heterogeneity in the impact of the minimum wage on employment across groups, with both positive and negative effects manifesting in the data" (p. 14) and that "the policy implication for the state government becomes less obvious" (p. 15). Keeping in mind the oversimplified assumptions in the basic supply and demand model, there seems to be more agreement regarding the effects of the minimum wage on certain specific subcategories of labour. It is clear that the labourers with low wages are the ones hardest hit, i.e., the ones close to the minimum wage enacted by law. Because of this, the most analysed subcategories of wage earners are teenagers, young adults, women, employees with a low level of education and skills, and different minority groups. One thesis on which researchers agree is that the minimum wage negatively affects young workers in particular. In the 1980s, there was a consensus that a ten percent increase in the minimum wage would cause a one to three percent increase in teenage unemployment [12] (p. 71). Brown et al.'s work was a particular important study in this sense at the time because it could be considered as a meta-analysis representative for the US economy. However, after 2000 there was a consistent wave of new studies in the American literature on the effects of minimum wage. A new meta-analysis done by Wolfson and Belman [13] (p. 2) claimed that the consensus range for the employment elasticity of the minimum wage shrank from [−0.3, 0.1] in the 80s and 90s to [−0.13, 0.07] at present. The relationship is still negative and statistically significantly, but closer to zero.

Jardim et al. [14] studied the impact of the minimum wage on low-wage employment in the State of Washington (USA), Seattle area, and observed that the increase of the minimum wage had a "sizable impact on jobs directly impacted by the increase and no cascading effects on other jobs under $19 per hour". Their results also confirmed that this change of the minimum wage was not only reducing low-wage employment but was also reducing the number of working hours. Moreover, the results suggested that the working hours decreased more than the minimum wage increased, especially for low-wage jobs. Belman and Wolfson [15] reached some very interesting conclusions after they studied the existing empirical studies on this important controversy: the minimum wage increase has a higher negative impact on the hours of work than on the number of jobs or employment rate and significant changes in the structure of employment (p. 109). They also concluded that: A minimum wage increase has a very limited impact on poverty (p. 336); It reduces the profitability of the respective companies (p. 394); There are very few studies on the impact of the minimum wage on output (p. 392); It has a positive impact on market prices by increasing operating costs (p. 384). This comprehensive study is also relevant for the screening of the methodologies used in various papers measuring the effects of the

minimum wage, many of them including panel data regressions, weighted least squares, and national data (p. 202, p. 206, p. 209, p. 214).

The problem with the lack of consensus on the effects of minimum wages can also be found in Europe. While most of the empirical work regarding minimum wages was done in the US, there are also studies that have used data from European countries. For example, Sturn [16] (p. 27) suggested that, in his study of 19 OECD countries, he found "little evidence for substantial dis-employment effects of minimum wages in the samples of low-skilled and youth labour market outcomes". This applies to low-skilled workers, female low-skilled workers, and young adults. Marimpi and Koning [17] (p. 1) argued that countries with a lower minimum wage for workers under the age of 25 have a 10 to 12 percent higher employment rate than countries with a uniform minimum wage. Furthermore, they claimed that the results "in countries with [lower] youth minimum wages are close to those in countries without minimum wages at all". On the other hand, Christl et al. [18] suggested that the effects of minimum wages in Europe are non-linear. This would mean that, if minimum wage rates are small, they have no negative or even a slightly positive effect, while high minimum wages have a clear negative effect on employment. They further claimed that countries such has Belgium or France, given their high minimum wages and strongly regulated labour markets, currently experience negative effects while East European countries could still benefit from an increase in minimum wages. Holtemöller and Pohle [19] focused on another effect of the minimum wage, the mitigation of inequality, in their analysis of Germany, a country that recently introduced it. Their findings confirmed the robust negative impact of the increase of the minimum wage on marginal employment and on the number of jobs. They found a positive impact on regular employment and the results led to the conclusion that there is no evidence to confirm the transformation of marginal employment into a regular one. They also used panel data regression on the state-industry level. The spillover effect (long-term) generated by the minimum wage increase is another aspect of the problem. According to Zhao and Sun's [20] findings, the increase of the minimum wage has a direct positive impact on the workers' efforts during their working hours. The spillover effect due to the minimum wage is confirmed by the results. Moreover, their findings suggest that the unemployment rate increases due to the minimum wage increase and indicates a moderate transfer of labour cost from the fired persons to the low-wage employees that are still keeping their job. In most of the countries, the minimum wage is considered a reliable public policy to enhance the bargaining power of the employees on the labour market, without significant harmful effects on them. However, when analysed as a price floor for labour, minimum wage could eliminate from the market some workers with lower productivity or who are perceived as having such. In this approach, minimum wage regulation is expected to have negative effects on employment in general and on some specific workers, especially in the long run [21,22]. From a theoretical point of view, this negative effect is higher in the less developed economies that are dominated by sectors less intensive in capital, with limited access to finance and with less sophisticated and developed financial industry [23–25]. Minimum wage is claimed to protect worker wealth and to reduce social inequalities but imposing it has the opposite effect on employers, especially during economic recessions. The increase of minimum wage also reduces access to the labour market for unskilled persons, less-educated persons, and teenagers (with less working experience) [26,27]. The gender inequality problem can also be aggravated when the minimum wage is significantly set above the equilibrium wage [28,29]. Public and private sectors are differently affected by minimum wage: while the public sector is quite immune to the problem of minimum wage, private businesses are seriously harmed [30]. Moreover, the use of minimum wage cannot be justified by economic cycles, either: during economic expansion it is redundant (because the demand for labour is high), and during economic crises, it is a barrier against economic recovery [31–33]. However, as in the case of the US, there are also a number of recent, well-documented studies which point towards a new consensus. One such example is Dube [7] (p. 50), wherein after accounting

for a considerable body of international evidence from the UK, Germany, Hungary, and the US, the author claims that the effects of an increase in the minimum wage on employment are very muted.

The situation in the case of the minimum wage is complicated even further, at least according to Andrews & Kasy [34], who claim that papers on this topic are affected by publication bias. In their view, editors are more likely to publish articles which find a negative (vis-à-vis a positive) and a statistically significant (vis-à-vis insignificant) relationship between minimum wages and employment. This alone does not imply that researchers are purposefully adjusting their models to skew results, but that the studies which by chance find a large negative correlation are relatively easier to publish Dube [7] (p. 40). Marotta and Greene [35] provided a more narrow study, limited to local increases in minimum wage above the federal/national level in 29 states (plus DC) and 42 localities. Their conclusion was the opposite to that of Neumark and Wascher: "… debate continues about the employment impacts of minimum wage increases, although the evidence points to economically insubstantial or no misemployment effects related to modest increases in the minimum wage". Meer and West [36] studied the impact of the lagged log dynamics of the minimum wage on employment in the United States, using static and dynamic panel data frameworks, and clearly concluded that: "the minimum wage negatively affects employment and that this occurs over a period of several years. The results from the distributed lag specification in first differences suggest that a 10% permanent increase in the real minimum wage reduces employment by about 0.7 percent after three years". Their study also confirmed the same outcome when the government was targeting the change of the real minimum wage (indexed with inflation) and not the nominal value. Our study used a similar methodology, by lagging the minimum wage change (for the long-term effect) and by using the dynamics of minimum wage in real terms.

To conclude, the literature review revealed a relative majority of empirical studies leading towards a negative and small impact of an increase in minimum wages on employment. This situation can be regarded as a new consensus. However, there is still considerable controversy with regard to the most sensitive social categories such as low skilled and low income marginal employees, female workers, the youth, and elderly people. The long-term and spillover effects were also confirmed by studies on the subject. The minimum wage remains one of the most (politically) important but still (economically) controversial acts of state intervention in the labour market.

## 3. Research Design

### 3.1. Research Hypotheses

Based on the existing theoretical background, our research is focused on the relationship between the dynamics of minimum wage and the dynamics of employment on the labour market. Considering the minimum wage to have the same economic effects as any imposed minimum price, and considering labour similar to any other tradable commodity, we propose the following research hypotheses:

**Hypothesis 1 (H1).** *The increase of the minimum wage is negatively impacting the dynamics of total employment: when the minimum wage is increased, labour becomes more expensive and employers are reluctant to hire more people;*

**Hypothesis 2 (H2).** *The increase of minimum wage is negatively influencing the dynamics of employment for younger people: younger people entering the labour market have limited working experience and, sometimes, their education constitutes a significant operating cost for the business;*

**Hypothesis 3 (H3).** *The increase of minimum wage is negatively correlated with the employment of aged people: aged persons tend to become less productive in low-skilled jobs requiring physical strength; aged people could imply a higher cost for the employer due to higher frequency of getting ill; aged people could be less adaptive to new technologies and very conservative regarding change*

*in general, being first on the firing list in case of an increase in minimum wages (that could also pressure higher wages and jobs if the increase is discoordinated with labour productivity dynamics);*

**Hypothesis 4 (H4).** *The increase of the minimum wage is negatively influencing the dynamics of employment of female workers: minimum wage is unfavourable for the most sensitive categories of workers, including female workers. The natural differrences in skills between men and women manifests itself in many jobs. The different negative impact of minimum wage laws on female and male employment is somewhat similar to the gender pay gap.*

**Hypothesis 5 (H5).** *The dis-employment effect of the minimum wage is exceeding one year. Wage is a market price influenced by labour market supply and demand conditions. When the state significantly interferes with market powers and alters this price too much, the dis-employment effect becomes a medium term, longer lasting, phenomenon. This happens especially when the minimum wage indexation formula is not correlated with labour productivity, is arbitrary, or is discretionary.*

### 3.2. Data Sample and Data Sources

The empirical research is focused on a panel of 22 countries, most of them from the European Union (only two are located outside EU—Australia and Turkey): Australia, Belgium, Bulgaria, the Czech Republic, Estonia, Ireland, Greece, Spain, France, Latvia, Lithuania, Luxemburg, Hungary, Malta, The Netherlands, Poland, Portugal, Romania, Slovenia, Slovakia, the United Kingdom and Turkey. This limitation of our data panel is due to the fact that we eliminated countries that have no minimum wage and those with no complete data on the selected variables.

We used annual data covering 18 years (1999–2016), for a total of 340 country–year observations. Therefore, the data panel is "balanced" (data covers all countries), is "fixed" (data covers all years), and is "long" (more countries—22, less years—18). The source of the data was the Department of Statistics of the International Labour Office. The real minimum wage is measured in constant 2011 USD, PPP adjusted. Descriptive statistics are presented in Table 1.

**Table 1.** Descriptive statistics for the variables included in the model.

| Statistics | LOG_MINWAG | LOG_PROD | LOG_LABOR | LOG_TOTEMPL | LOG_AGE_15_24 | LOG_AGE_45_54 |
|---|---|---|---|---|---|---|
| Mean | 0.0237 | 0.0086 | 0.0027 | 0.0027 | −0.0096 | 0.0050 |
| Median | 0.0180 | 0.0070 | 0.0031 | 0.0043 | −0.0055 | 0.0065 |
| Maximum | 0.1808 | 0.0694 | 0.0923 | 0.0431 | 0.1249 | 0.0514 |
| Minimum | −0.0902 | −0.0475 | −0.1161 | −0.0647 | −0.1468 | −0.0512 |
| Std. Dev. | 0.0270 | 0.0135 | 0.0130 | 0.0120 | 0.0313 | 0.0130 |
| Skewness | 1.5833 | 0.2683 | −2.1298 | −1.4103 | −0.5085 | −0.7152 |
| Kurtosis | 10.73 | 5.08 | 33.63 | 8.37 | 5.48 | 5.48 |
| **Statistics** | **LOG_AGE_55_64** | **LOG_FEM_TOT** | **LOG_FEM_15_24** | **LOG_FEM_45_54** | | **LOG_FEM_55_64** |
| Mean | 0.0174 | 0.0044 | −0.0095 | 0.0076 | | 0.0236 |
| Median | 0.0187 | 0.0053 | −0.0049 | 0.0068 | | 0.0209 |
| Maximum | 0.1359 | 0.0635 | 0.1549 | 0.0872 | | 0.1761 |
| Minimum | −0.1238 | −0.0672 | −0.1231 | −0.0718 | | −0.1761 |
| Std. Dev. | 0.0239 | 0.0132 | 0.0350 | 0.0174 | | 0.0358 |
| Skewness | −0.4783 | −0.6406 | 0.0412 | 0.2173 | | −0.1111 |
| Kurtosis | 7.92 | 7.54 | 4.73 | 6.34 | | 8.88 |

Source: Own estimations based on ILO Data.

### 3.3. Variables, Model's Equations and Data Sample

The model that we proposed to be tested through the use of panel data analysis used the following explained variables (these variables were selected based on the proposed research hypotheses):

(1) LOG_TOTALEMPL—the logarithmic values of the yearly change for the total employment—$TOTALEMPL_1/TOTALEMPL_0$;

(2) LOG_AGE_15_24—the logarithmic values of the yearly change for the total employment of persons aged between 15 and 24 years—$AGE\_15\_24_1/AGE\_15\_24_0$;

(3) LOG_AGE_45_54—the logarithmic values of the yearly change for the total employment of persons aged between 45 and 55 years—$AGE\_45\_55_1/AGE\_45\_55_0$;

(4) LOG_AGE_55_64—the logarithmic values of the yearly change for the total employment of persons aged between 55 and 64 years—$AGE\_55\_64_1/AGE\_55\_64_0$;

(5) LOG_FEM_TOTAL—the logarithmic values of the yearly change for the total female employment—$FEM\_TOTAL_1/FEM\_TOTAL_0$

(6) LOG_FEM_15_24—the logarithmic values of the yearly change for the total female employees aged between 15 and 24 years—$FEM\_15\_24_1/FEM\_15\_24_0$;

(7) LOG_FEM_45_54—the logarithmic values of the yearly change for the total female employees aged between 45 and 55 years—$FEM\_45\_55_1/FEM\_45\_55_0$;

(8) LOG_FEM_55_64—the logarithmic values of the yearly change for the total female employees aged between 55 and 64 years.

The explanatory variable in our equations is the logarithmic value of the dynamic of minimum wage—LOG_MINWAGE (constant 2011 USD, PPP standard). The model uses two exogenous controlling variables: the logarithmic values of the yearly change of the labour productivity—LOG_PROD and the logarithmic values of the yearly change of the total labour—LOG_LABOR. In Table 2, we presented a short description of the variables and their data sources.

**Table 2.** Descriptive statistics for the variables included in the model.

| A. Explanatory variable | Symbol | Definitions and Data source |
|---|---|---|
| Real minimum wage | MINWAGE | Minimum wage, expressed in constant 2011 USD, PPP standards, yearly data, ILO Statistics |
| **B. Controlling variables** | **Symbol** | **Definitions and Data source** |
| Labour productivity | PROD | Labour productivity, yearly data, ILO Statistics. |
| Labour force | LABOR | Labour force, yearly data, ILO Statistics. In this category are all people aged 15 and older (currently employed and unemployed people but seeking for a job and first-time job-seekers). |
| **C. Explained variables** | **Symbol** | **Definitions and Data source** |
| Total employment | TOTALEMPL | Total employment, yearly data, ILO Statistics. |
| Total employment by age (age 15–24 years) | AGE_15_24 | Total employed persons by age, people aged between 15 and 24 years, yearly data, ILO Statistics. |
| Total employment by age (age 45–55 years) | AGE_45_54 | Total employed persons by age, people aged between 45 and 55 years, yearly data, ILO Statistics. |
| Total employment by age (age 55–64 years) | AGE_55-64 | Total employed persons by age, people aged between 55 and 65 years, yearly data, ILO Statistics. |
| Total employment by gender (female employment) | FEM_TOT | Total female employed persons, yearly data, ILO Statistics. |
| Total employment by age and sex (15–24 age, female employment) | FEM_15_24 | Total female employed persons by age, people aged between 15 and 24 years, yearly data, ILO Statistics. |
| Total employment by age and sex (15–24 age, female employment) | FEM_45_54 | Total female employed persons by age, log change, people aged between 45 and 55 years, yearly data, ILO Statistics. |
| Total employment by age and sex (15–24 age, female employment) | FEM_55_64 | Total female employed persons by age, log change, people aged between 55 and 65 years, yearly data, ILO Statistics. |

In this research, we used eight data panels corresponding to the eight selected categories of employed persons (total and differentiated by age and by gender). The equations of the basic model used to estimate the impact of the dynamics of the minimum wage on the dynamics of employment are the following:

*Panel 1 (the impact of the minimum wage dynamic on the total employment's dynamic):*

$$LOG\_TOTALEMPL_{it} = a_1 + b_1 \times LOG\_MINWAGE_{it} + c_1 \times LOG\_PROD_{it} + d_1 \times LOG\_LABOR_{it} + c + \varepsilon_{it} \quad (1)$$

*Panel 2 (the impact of the minimum wage dynamic on people aged between 15 and 24's employment dynamic):*

$$LOG\_AGE\_15\_24_{it} = a_2 + b_2 \times LOG\_MINWAGE_{it} + c_2 \times LOG\_PROD_{it} + d_2 \times LOG\_LABOR_{it} + c + \varepsilon_{it} \quad (2)$$

*Panel 3 (the impact of the minimum wage dynamic on people aged between 45 and 55's employment dynamic):*

$$LOG\_AGE\_45\_54_{it} = a_3 + b_3 \times LOG\_MINWAGE_{it} + c_3 \times LOG\_PROD_{it} + d_3 \times LOG\_LABOR_{it} + c + \varepsilon_{it} \quad (3)$$

*Panel 4 (the impact of the minimum wage dynamic on people aged between 55 and 64's employment dynamic):*

$$LOG\_AGE\_55\_64_{it} = a_4 + b_4 \times LOG\_MINWAGE_{it} + c_4 \times LOG\_PROD_{it} + d_4 \times LOG\_LABOR_{it} + c + \varepsilon_{it} \quad (4)$$

*Panel 5 (the impact of the minimum wage dynamic on the total female population's employment dynamic):*

$$LOG\_FEM\_TOT_{it} = a_5 + b_5 \times LOG\_MINWAGE_{it} + c_5 \times LOG\_PROD_{it} + d_5 \times LOG\_LABOR_{it} + c + \varepsilon_{it} \quad (5)$$

*Panel 6 (the impact of the minimum wage dynamic on the female population aged between 15 and 24's employment dynamic):*

$$LOG\_FEM\_15\_24_{it} = a_6 + b_6 \times LOG\_MINWAGE_{it} + c6 \times LOG\_PROD_{it} + d_6 \times LOG\_LABOR_{it} + c + \varepsilon_{it} \quad (6)$$

*Panel 7 (the impact of the minimum wage dynamic on the female population aged between 45 and 54's employment dynamic):*

$$LOG\_FEM\_45\_54_{it} = a_7 + b_7 \times LOG\_MINWAGE_{it} + c_7 \times LOG\_PROD_{it} + d_7 \times LOG\_LABOR_{it} + c + \varepsilon_{it} \quad (7)$$

*Panel 8 (the impact of the minimum wage dynamic on the female population aged between 55 and 64's employment dynamic):*

$$LOG\_FEM\_55\_64_{it} = a_8 + b_8 \times LOG\_MINWAGE_{it} + c_8 \times LOG\_PROD_{it} + d_8 \times LOG\_LABOR_{it} + c + \varepsilon_{it} \quad (8)$$

*Long-term impact is estimated by lagging minimum wage dynamic with k lags (all panels):*

$$LOG\_Y_{it} = a_9 + b_9 \times LOG\_MINWAGE_{i[t-k]} + c_9 \times LOG\_PROD_{it} + d_9 \times LOG\_LABOR_{it} + c + \varepsilon_{it} \quad (9)$$

where k = 1, 2, and 3 (lagging only LOG_MINWAGE) and LOG_Yit are all dependent variables tested by our models (LOG_TOTALEMPL, LOG_AGE_15_24, LOG_AGE_45_54, LOG_AGE_55_64, LOG_FEM_TOT, LOG_FEM_15_24, LOG_FEM_45_54, and LOG_FEM_55_64).

The steps in our panel data analysis presented are the following: (i) correlation matrix between all the variables; (ii) unit root tests for individual variables; (iii) cointegration tests for each data panel, VECM analysis for long-term relationships in each data panel, Wald tests for short-term relationships, estimators for basic model, fixed effects/random effects test, adjustment of the estimators with fixed effects/random effect, and estimating the long-term effect by lagging minimum wage dynamic with 1, 2, and 3 lags. For econometric analysis, we used Eviews 12.

### 3.4. Correlation Matrix

The correlation matrix is presented in Table 3. The results indicate a negative correlation between the dynamics of minimum wage and that of employment (for all

categories, without exception; higher minimum wage induces a lower level of employment).

**Table 3.** Descriptive statistics for the variables included in the model.

| | TOTEMPL | AGE_15_24 | AGE_45_55 | AGE_55_64 | FEM_TOTAL | FEM_15_24 | FEM_45_55 | FEM_55_64 | MINWAGE | LGPROD | LGLABOUR |
|---|---|---|---|---|---|---|---|---|---|---|---|
| **TOTEMPL** | 1.00 | | | | | | | | | | |
| **AGE_15_24** | 0.70 | 1.00 | | | | | | | | | |
| **AGE_45_55** | 0.61 | 0.36 | 1.00 | | | | | | | | |
| **AGE_55_64** | 0.55 | 0.29 | 0.18 | 1.00 | | | | | | | |
| **FEM_TOTAL** | 0.83 | 0.55 | 0.60 | 0.48 | 1.00 | | | | | | |
| **FEM_15_24** | 0.59 | 0.87 | 0.28 | 0.28 | 0.53 | 1.00 | | | | | |
| **FEM_45_55** | 0.56 | 0.23 | 0.72 | 0.20 | 0.65 | 0.23 | 1.00 | | | | |
| **FEM_55_64** | 0.39 | 0.16 | 0.11 | 0.76 | 0.44 | 0.16 | 0.19 | 1.00 | | | |
| **MINWAGE** | −0.09 | −0.07 | −0.09 | −0.04 | −0.11 | −0.08 | −0.12 | −0.02 | 1.00 | 0.13 | -0.08 |
| **LGPROD** | −0.10 | −0.07 | 0.00 | −0.10 | −0.20 | −0.15 | −0.11 | −0.07 | 0.13 | 1.00 | |
| **LGLABOUR** | 0.50 | 0.28 | 0.33 | 0.32 | 0.44 | 0.28 | 0.33 | 0.26 | −0.08 | −0.09 | 1.00 |

Source: Own estimations based on ILO Data.

Additionally, we can observe a positive correlation between the dynamics of minimum wages and the dynamics of labour productivity (higher minimum wage determines higher labour productivity) and a negative correlation between the dynamics of minimum wages and the dynamics of total labour force (higher minimum wage leads to lower available labour force). All these correlations' values confirm all the theoretical hypotheses proposed for our research.

*3.5. Unit-Root Tests*

The next step was to check if the variables included in our model are stationary or not. For this, we used the four recommended unit root tests for panel data: the Levin, Lin, and Chu test (2002); the Im, Pesaran, and Shin test (2003); the Lean and Smyth test (2010); the Wang and PP—Fisher Chi-square test (2011). The results of these tests are presented in Table 4.

**Table 4.** Unit root tests.

| Unit Root Tests | LOG_MINWAG | LOG_PROD | LOG_LABOR | LOG_TOTAL EMPL | LOG_AGE_15_24 | LOG_AGE_45_54 |
|---|---|---|---|---|---|---|
| Levin, Lin, and Chu t * | −7.841 | −5.727 | −5.276 | −4.068 | −1.836 | −3.053 |
| | 0.000 * | 0.000 | 0.000 | 0.000 | 0.033 | 0.001 |
| Im, Pesaran, and Shin W-stat | −6.757 | −4.687 | −7.178 | −3.654 | −3.397 | −3.784 |
| | 0.000 * | 0.000 | 0.000 | 0.000 | 0.000 | 0.000 |
| ADF—Fisher Chi-square | 125.431 | 94.772 | 137.959 | 79.001 | 81.847 | 82.568 |
| | 0.000 | 0.000 | 0.000 | 0.001 | 0.001 | 0.000 |
| PP—Fisher Chi-square | 190.228 | 166.555 | 265.577 | 123.871 | 139.332 | 131.924 |
| | 0.000 | 0.000 | 0.000 | 0.000 | 0.000 | 0.000 |
| **Unit Root Tests** | **LOG_AGE_55_64** | **LOG_FEM_TOTAL** | | **LOG_FEM_15_24** | **LOG_FEM_45_54** | **LOG_FEM_55_64** |
| Levin, Lin, and Chu t * | −3.340 | −2.352 | | −3.086 | −3.222 | −4.317 |
| | 0.000 | 0.009 | | 0.001 | 0.001 | 0.000 |
| Im, Pesaran, and Shin W-stat | −4.242 | −3.056 | | −3.762 | −3.029 | −5.465 |
| | 0.000 | 0.001 | | 0.000 | 0.001 | 0.000 |
| ADF—Fisher Chi-square | 92.695 | 71.566 | | 84.898 | 72.114 | 106.851 |
| | 0.000 | 0.005 | | 0.000 | 0.005 | 0.000 |
| PP—Fisher Chi-square | 220.088 | 153.012 | | 167.535 | 129.562 | 233.744 |
| | 0.000 | 0.000 | | 0.000 | 0.000 | 0.000 |

Source: Own estimations based on ILO Data, *—5% confidence level.

The results of the unit root tests significantly rejected the null hypothesis of the presence of a unit root for all the variables included in our model. Therefore, we can conclude that all the series are stationary (all the data are already log differentiated values of the variables). All the data series are time invariant, having a mean, variance, and covariance constant over time. Therefore, no other data transformation is needed for the next steps of the analysis.

### 3.6. Cointegration Tests

Our research proposes eight panels with eight sets of variables corresponding to the various employment categories considered (total employment, employment by age, employment by gender, employment by gender and age). Thus, we performed eight cointegration tests for each panel. We used recommended tests for panel data analysis, namely the Pedroni residual cointegration test (2004), the Kao cointegration test (1999), and the Fisher/Johansen combined cointegration test (1988). The results of these cointegration tests are presented in Table 5. According to these results, we can conclude that the vectors of each panel are cointegrated for all models and panels (the null hypothesis of "no cointegration/no deterministic trend" is not rejected at a conventional size of 0.05). These results suggest a possible long-run equilibrium and a common stochastic trend for all tested data panels.

**Table 5.** Cointegration tests' results for the panels included in the analysis.

| Panels: | Panel 1 | Panel 2 | Panel 3 | Panel 4 | Panel 5 | Panel 6 | Panel 7 | Panel 8 |
|---|---|---|---|---|---|---|---|---|
| **Pedroni Residual Cointegration Test (H0: no cointegration/no deterministic trend)** | | | | | | | | |
| Panel v-Statistic | 0.817 | 0.991 | 0.973 | 0.968 | 0.972 | 0.970 | 0.968 | 0.991 |
| Panel rho-Statistic | 0.314 | 0.228 | 0.335 | 0.247 | 0.331 | 0.183 | 0.247 | 0.321 |
| Panel PP-Statistic | 0.000 | 0.000 | 0.000 | 0.000 | 0.000 | 0.000 | 0.000 | 0.000 |
| Panel ADF-Statistic | 0.000 | 0.000 | 0.000 | 0.000 | 0.000 | 0.000 | 0.000 | 0.000 |
| Panel v-Stat. (weighted) | 0.999 | 0.999 | 0.998 | 0.999 | 0.999 | 0.999 | 0.999 | 0.998 |
| Panel rho-Stat. (weighted) | 0.315 | 0.296 | 0.281 | 0.295 | 0.293 | 0.316 | 0.295 | 0.382 |
| Panel PP-Statistic | 0.000 | 0.000 | 0.000 | 0.000 | 0.000 | 0.000 | 0.000 | 0.000 |
| Panel ADF-Statistic | 0.000 | 0.000 | 0.000 | 0.004 | 0.001 | 0.001 | 0.004 | 0.004 |
| **Kao Cointegration Test (H0: no cointegration/no deterministic trend)** | | | | | | | | |
| ADF | 0.025 | 0.001 | 0.039 | 0.000 | 0.001 | 0.002 | 0.1929 | 0.000 |
| **Fisher Cointegration test (H0: no deterministic trend)** | | | | | | | | |
| Fisher stat (none) | 0.000 | 0.000 | 0.000 | 0.000 | 0.000 | 0.000 | 0.000 | 0.000 |
| Presence of Panel Cointegration | Yes | Yes | Yes | Yes | Yes | Yes | Yes | Yes |

Source: own estimations based on panel data.

### 3.7. VECM Tests (Long-Run Relationship)

The next step was to further test this long-term relationship by using the Vector Error Correction Model (VECM). The synthetic results of the VECM test are presented in Table 6.

**Table 6.** VECM Results for each panel: Error correction terms (ECT) statistical significance.

| Panels: | Coefficient | Std. Error | *t*-Statistic | Prob. |
|---|---|---|---|---|
| ECT (Panel 1) | −0.242271 | 0.087925 | −2.755.433 | 0.0060 * |
| ECT (Panel 2) | −0.058175 | 0.057313 | −1015030 | 0.3103 *** |
| ECT (Panel 3) | −0.117459 | 0.061002 | −1.925.482 | 0.0544 * |
| ECT (Panel 4) | 0.004637 | 0.020159 | 0.230033 | 0.8181 |
| ECT (Panel 5) | −0.249458 | 0.096972 | −2572468 | 0.0102 * |
| ECT (Panel 6) | −0.116572 | 0.069189 | −1684821 | 0.0923 ** |
| ECT (Panel 7) | −0.149247 | 0.073967 | −2017760 | 0.0438 * |
| ECT (Panel 8) | −0.007027 | 0.015317 | −0.458760 | 0.6465 |

Source: Own estimations based on panel data; ECT—Error Correction Term—C(0); */**/*** is the statistical relevance of each coefficient—*—1% confidence level, **—5% confidence level, ***—10% confidence level.

VECM tests confirmed for almost all panels the existence of a long term relationship between dependent variables (the log dynamic of minimum wage, log dynamic of labour productivity, and log dynamic of total labour force), except for LOG_EMPL_55_64 (aged employment, Panel 4) and LOG_FEM_55_64 (aged female employment, Panel 8), where the long-run determination signaled by Error Correction Term was not statistically significant.

### *3.8. Wald Panel's Test (Short-Run Relationship)*

To test the short-run relationship between the variables included in each panel, we used the Wald test. The results of this test are summarized in Table 7. As we observed from the results, with few exceptions (LOG_TOTEMPL—Panel 1 and LOG_AGE_45_54—Panel 3), the short-run relationship was not confirmed by this test.

**Table 7.** Wald test results (short-run relationship).

| Chi-Square Wald Test | Value | Probability |
|---|---|---|
| Panel 1 (LOG_TOTEMPL dependent variable)) | 7.344.401 | 0.0254 * |
| Panel 2 (LOG_AGE_15_24 dependent variable) | 1.591.887 | 0.4512 |
| Panel 3 (LOG_45_54 dependent variable) | 1.105.620 | 0.0040 * |
| Panel 4 (LOG_55_64 dependent variable) | 2.830.863 | 0.2428 |
| Panel 5 (LOG_FEM_TOT dependent variable) | 1.631.577 | 0.4423 |
| Panel 6 (LOG_FEM_15_24 dependent variable) | 0.050214 | 0.9752 |
| Panel 7 (LOG_FEM_45_55 dependent variable) | 1.410.654 | 0.4939 |
| Panel 8 (LOG_FEM_55_64 dependent variable) | 4.599.356 | 0.1003 |

Source: Own estimations based on panel data, *—5% confidence level.

## 4. Results and Discussions

The main results of the basic model are presented in Table 8. Except for the category of female employed persons aged between 55 and 64 years, all models confirmed a negative relationship between the log change (dynamics) of the minimum wage and the log change (dynamics) of employment rate. These results practically confirmed that an increase of the minimum wage destroys jobs and reduces employment (including categories by age and gender with the already mentioned exception).

**Table 8.** Main results from the basic model.

| Variables | Eq 1 | Eq 2 | Eq 3 | Eq 4 | Eq 5 | Eq 6 | Eq 7 | Eq 8 |
|---|---|---|---|---|---|---|---|---|
| LOG_MINWAG | −0.020 | −0.051 | −0.03 *** | −0.002 | −0.029 | −0.0539 | −0.054 *** | 0.007 |
| | −0.973 | −0.878 | −1.448 | −0.050 | −1.287 | −0.834 | −1.709 | 0.102 |
| LOG_LABOR | 0.452 ** | 0.650 ** | 0.332 ** | 0.578 ** | 0.424 * | 0.710 ** | 0.426 ** | 0.709 ** |
| | 10.834 | 5.381 | 6.765 | 6.352 | 9.030 | 5.305 | 6.491 | 5.093 |
| LOG_PROD | −0.046 | −0.095 | 0.040 | −0.120 | −0.146 * | −0.318 * | −0.095 *** | −0.124 |
| | −1.134 | −0.806 | 0.837 | −1.364 | −3.197 | −2.451 | −1.487 | −0.916 |
| C | 0.002 * | −0.009 * | 0.005 * | 0.017 * | 0.005 * | −0.007 * | 0.009 * | 0.023 * |
| | 2.965 | −4.077 | 4.948 | 9.869 | 5.858 | −2.905 | 6.907 | 8.580 |
| Adj. R-sq. | 0.040 | 0.335 | 0.055 | 0.100 | 0.212 | 0.088 | 0.117 | 0.063 |
| F-Statistic | 41.885 | 10.871 | 16.503 | 14.834 | 34.443 | 13.048 | 17.445 | 9.308 |
| DW Test | 1.381 | 1.440 | 1.485 | 1.692 | 1.567 | 1.680 | 1.591 | 1.941 |

Source: Own estimations, first value is coefficient, second value is t-statistic, */**/*** is the statistical relevance of each coefficient—*—1% confidence level, **—5% confidence level, ***—10% confidence level; Eq 1, Eq 2… stand for Equation (1), Equation (2)…

　　　　Regarding the controlling variables, the results also indicated a logical negative relationship between the log change of the employment rate and the log change of the labour productivity (the exception is the category of employed persons aged 45–55 years, for which we found a positive relationship). This means that any increase in the labour productivity decreases the employment rate. Additionally, the results confirmed the logical positive relationship between the dynamic (log change) of total supply with labour and the dynamic (log change) of employment (when the labour force is increasing, the employment increases for all categories of employed persons included in the study).

### 4.1. Random Effects/Fixed Effects Adjustments

　　　　One of the key issues in the panel data analysis is the presumable presence of the fixed and random effects (variables can change over time or countries). Therefore, the next step is to check if our estimates are not biased to the fixed effects (there are omitted variables that are not changing over time but rather varying over the countries in the data panel, such as in education level or, in our case, in the size of the country) and to the random effects (there are omitted variables that are not changing over countries but varying over time, such as labour market-specific indicators or labour market regulation, in our case). In some panels, both effects can be simultaneously present, biasing the estimates of the model. The recommended tests, in this case, are: the Hausman test (for random effects) and the redundant fixed effects test (for fixed effects).

　　　　The results of the Hausman tests (cross-section and period) are presented in Table 9. As we observed, the null hypothesis of random effects cannot be rejected in all cases. In the case of Panel 4 (LOG_AGE_55_64), Panel 6 (LOG_FEM_15_24), and Panel 8 (LOG_FEM_55_64), the random effects suggested by the tests were used to correct in the estimators. Otherwise, random effects (cross-section, period) were not valid.

**Table 9.** Hausman test results (random effects).

| Test Summary | Chi-Sq. Statistic | Prob. |
|---|---|---|
| Panel (LOG_TOTEMPL dependent variable) | | |
| Cross-section random | 11.543 | 0.009 * |
| Period random | 6.255 | 0.100 * |
| Panel 2 (LOG_AGE_15_24 dependent variable) | | |
| Cross-section random | 6.376 | 0.095 ** |
| Period random | 5.385 | 0.146 *** |
| Panle 3 (LOG_AGE_45_54 dependent variable) | | |
| Cross-section random | 10.868 | 0.013 * |
| Period random | 18.020 | 0.000 * |
| Panel 4 (LOG_AGE_55_64 dependent variable) | | |
| Cross-section random | 1.682 | 0.641 |
| Period random | 2.575 | 0.462 |
| Panel 5 (LOG_FEM_TOT dependent variable) | | |
| Cross-section random | 13.685 | 0.003 * |
| Period random | 6.460 | 0.091 * |
| Panel 6 (LOG_FEM_15_24 dependent variable) | | |
| Cross-section random | 3.135 | 0.371 |
| Period random | 3.818 | 0.282 |
| Panel 7 (LOG_FEM_45_54 depedent variable) | | |
| Cross-section random | 12.663 | 0.005 * |
| Period random | 10.740 | 0.013 * |
| Panel 8 (LOG_FEM_55_64 dependent variable) | | |
| Cross-section random | 0.971 | 0.808 |
| Period random | 1.948 | 0.583 |

Source: Our estimations based on panel data, *—confidence level 1%, **—confidence level 5%, ***—confidence level 10%.

To have a more conclusive perspective, we continued the analysis by also running the redundant fixed effects test on the data panels. The results are summarized in Table 10.

**Table 10.** Redundant fixed effects test.

| Panels | Cross-Section F | Cross-Section Chi-Square | Period F | Period Chi-Square | Cross-Section/Period F | Cross-Section/Period Chi-Square |
|---|---|---|---|---|---|---|
| Panel 1 (dependent variable LOG_TOTEMPL) | | | | | | |
| Statistic | 0.952 | 21.815 | 6.708 | 104.485 | 3.486 | 122.449 |
| Prob | 0.523 | 0.410 | 0.000 | 0.000 | 0.000 | 0.000 |
| Panel 2 (dependent variable LOG_AGE_15_24) | | | | | | |
| Statistic | 1.921 | 42.759 | 6.236 | 98.016 | 3.711 | 129.116 |
| Prob | 0.009 | 0.003 | 0.000 | 0.000 | 0.000 | 0.000 |
| Panel 3 (dependent variable LOG_AGE_45_54) | | | | | | |
| Statistic | 2.484 | 54.423 | 2.562 | 43.419 | 2.711 | 98.483 |
| Prob | 0.000 | 0.000 | 0.001 | 0.000 | 0.000 | 0.000 |
| Panel 4 (dependent variable LOG_AGE_55_64) | | | | | | |
| Statistic | 2.557 | 55.916 | 2.519 | 42.725 | 2.562 | 93.684 |
| Prob | 0.000 | 0.000 | 0.001 | 0.000 | 0.000 | 0.000 |
| Panel 5 (dependent variable LOG_FEM_TOT) | | | | | | |
| Statistic | 1.590 | 35.731 | 3.718 | 61.476 | 2.579 | 94.244 |
| Prob | 0.050 | 0.023 | 0.000 | 0.000 | 0.000 | 0.000 |
| Panel 6 (dependent variable LOG_FEM_15_24) | | | | | | |
| Statistic | 1.558 | 35.047 | 4.873 | 78.684 | 2.940 | 105.704 |
| Prob | 0.057 | 0.028 | 0.000 | 0.000 | 0.000 | 0.000 |

| | Panel 7 (dependent variable LOG_FEM_45_54) | | | | | |
|---|---|---|---|---|---|---|
| Statistic | 2.229 | 49.188 | 1.809 | 31.177 | 2.255 | 83.632 |
| Prob | 0.002 | 0.001 | 0.029 | 0.013 | 0.000 | 0.000 |
| | Panel 8 (dependent variable LOG_FEM_55_64) | | | | | |
| Statistic | 2.326 | 51.200 | 1.710 | 29.527 | 2.076 | 77.622 |
| Prob | 0.001 | 0.000 | 0.043 | 0.021 | 0.000 | 0.000 |

Source: Our estimations based on panel data.

According to the results of this fixed effects test, the presence of fixed effects is valid for all the panels, with no exceptions (both for cross-section and period fixed effects).

### 4.2. Fitted Estimators with Random Effects/Fixed Effects

In the next stage of our analysis, we fitted the estimators of the basic model (see Table 8) with the fixed effects and with the random effects accordingly with the results of the Hausman test and redundant fixed effects test (Tables 9 and 10). In Table 11, we presented the best fitted estimators that we found by successively applying these effects on the panels' regressions. The estimators are clearly improved and possible biases eliminated.

The results reconfirmed the negative relationship between the log change (dynamic) of the minimum wage and the log change (dynamic) of employment. This negative relationship is also confirmed for all considered categories of employment (by age and female gender) with one exception—the aged employment dynamic is positively influenced by the dynamic of the minimum wage. The possible explanation of this result for the aged persons could be the following: when the minimum wage increases, the disemployment of teenagers/female persons is compensated by the aged persons that can be preferred by employers due to their accumulated experience. However, this is an interesting result that should be further investigated in our subsequent research.

**Table 11.** The best fitted estimators (by taking into consideration fixed/random effects).

| Variables | Panel 1 | Panel 2 | Panel 3 | Panel 4 | Panel 5 | Panel 6 | Panel 7 | Panel 8 |
|---|---|---|---|---|---|---|---|---|
| LOG_MINWAG | −0.018 * | −0.031 * | −0.048 * | 0.028 * | −0.033 * | −0.051 * | −0.076 * | 0.044 * |
| | −4.84 | −3.42 | −7.79 | 2.29 | −11.41 | −2.82 | −6.56 | 3.36 |
| LOG_LABOR | 0.453 * | 0.551 * | 0.302 * | 0.596 * | 0.406 * | 0.653 * | 0.382 * | 0.711 * |
| | 26.66 | 14.651 | 19.82 | 22.36 | 57.93 | 16.72 | 16.15 | 33.84 |
| LOG_GPROD | −0.153 * | −0.424 * | −0.035 * | −0.225 * | −0.245 * | −0.491 * | −0.129 * | −0.196 * |
| | −12.45 | −12.41 | −1.86 | −8.63 | −21.89 | −10.61 | −3.93 | −6.57 |
| C | 0.003 * | −0.007 * | 0.006 * | 0.017 * | 0.006 * | −0.006 * | 0.009 * | 0.022 * |
| | 12.60 | −8.27 | 14.46 | 31.90 | 31.15 | −5.69 | 15.38 | 19.73 |
| Adj. R-sq. | 0.831 | 0.629 | 0.689 | 0.657 | 0.916 | 0.617 | 0.591 | 0.588 |
| F-Statistic | 97.291 | 34.270 | 44.554 | 38.675 | 214.798 | 32.654 | 29.351 | 29.068 |
| DW Test | 2.034 | 1.959 | 1.969 | 2.030 | 2.006 | 2.000 | 1.953 | 2.010 |
| **Random effects/Fixed effects fitting parameters:** | | | | | | | | |
| **GLS Weights** | Period SUR | Period SUR | Period SUR | Period SUR | Period SUR | Period SUR | Period SUR | Period SUR |
| **Unbalanced SUR** | No | No | Yes | Yes | No | No | No | Yes |
| **Coefficient covariance method** | White (diagonal) | White period | Period SUR (PCSE) | White period | White (diagonal) | White (diagonal) | Ordinary | White period |

Source: Own estimations. The first value is the coefficient and second value is the t-statistic; *—5% confidence level.

The fitted estimators also confirmed the negative relationship with the labour's productivity (LOG_PROD) and a positive relationship with the labour supply dynamic (LOG_LABOR) for all considered employment categories.

*4.3. The Long-Run Dis-Employment Effect of the Minimum Wage Dynamic*

The VECM indicated a long-run impact of the log change of the minimum wage on employment (various categories). The final step is to identify the long-run impact of the log change of the minimum wage on total employment dynamics. We used lags of 1, 2, and 3 for the minimum wage for estimating this long-run effect. The results are summarized in Table 12. We kept the same fitting parameters for the panel regressions for all lags.

**Table 12.** The impact of lagged minimum wage's dynamic on the total employment's change.

| Lags: | Coefficient | Std. Error | *t*-Statistic | Prob. | R-Squared | F-Stat |
|---|---|---|---|---|---|---|
| LOG_MINWAG(-1) | −0.020943 * | 0.003939 | −5,3163 | 0.0000 | 0.867661 | 121,2923 |
| LOG_MINWAG(-2) | −0.021626 * | 0.005701 | −3,7933 | 0.0002 | 0.826276 | 87,2911 |
| LOG_MINWAG(-3) | 0.002238 | 0.004358 | 0.513574 | 0.6079 | 0.844158 | 98,5176 |
| LOG_MINWAG(-4) | 0.022330 | 0.017003 | 1313254 | 0.1902 | 0.596803 | 26,64324 |

Source: Own estimations based on ILO data, *—1% significance level.

As the results suggest, the long-run influence of the minimum wage dynamics on total employment dynamics lasts at least 2–3 years. After that, this relationship becomes statistically insignificant.

**5. Conclusions**

The minimum wage is one of the most sensitive and debated public policies today. The state intervention on the labour market by imposing on the employers a minimum price when they hire labour for their businesses is commonly seen to have a positive effect on the wealth of the employed persons by strengthening their bargaining power. However, the obligation to pay a minimum wage can also be seen as a barrier to the development of a business and to the employment of labour, especially less skilled or less educated persons, female workers (due to their limited physical abilities), aged persons (due to their decreasing physical abilities), or younger people (due to their lack of experience). Therefore, the increase of the minimum wage without a corresponding increase in the market capacity to pay for this imposed higher price is destined to have a negative, rather than a positive, impact.

This study confirmed almost all tested research hypotheses: the increase of the minimum wage is negatively influencing the dynamics of total employment, is negatively influencing the dynamics of the employment of the younger people, is negatively correlated with the employment of female workers, and this impact is resilient on a long-term basis (the change of the minimum wage in the past is still producing a negative impact on the current employment rate). The results of this study did not confirm the research hypothesis relating to the dynamics of the minimum wage and the dynamics of the employment rate for aged persons (55–64 years). Our explanations, in this case, are related to the fact that the countries included in our panel are mainly from EU, are developed and have a highly skilled and educated labour force (physical abilities are less important in this case). The majority of employed persons in this category are not paid the minimum wage and the high experience of this aged group is worth more than the increasing cost induced by the increase of the minimum wage. This positive impact is present not only at the level of total employment but also at the level of total female employment (aged 55–64). These results challenged us and further analysis should include more data about the countries in the panel, especially for this category of age, in order to confirm all the suppositions that could explain the results, which were the opposite of those expected.

The findings of this study are consistent with other recent empirical studies. The identified—statistically relevant—negative impact on the dynamics of the employment rate (including its long-running identified effects) invites, at least, a reconsideration of the

economic effectiveness of this public policy, especially in the case of developed countries such as those included in this study. As other studies have illustrated, the most sensitive categories of the labour market (such as the youth, the less skilled or less educated, and female workers) can be significantly affected by losing their jobs or by remaining unemployed for a longer period of time. Adjustments with the fixed effects/random effects improved the statistical relevance of the models and coefficients and were also consistent with previous similar studies.

This research has some limitations derived from the fact that we were interested in having a complete data panel. Therefore, we excluded the countries with incomplete observations for the period included in the analysis. Finally, our complete panel is quite limited to EU countries (most of them being from this area). This is a limitation that we intend to improve in our further studies, which will use incomplete panel data and with which we will compare the results. Moreover, we believe that the panel could be representative for the global market, and that the results could be extrapolated with minor errors due to the economic importance of these countries. In the future, we intend to include more dummy variables that will take into consideration the size of the countries, their economic development, and their economic structure (labour intensive versus capital intensive productive sectors). Additionally, we intend to check the effect that runs opposite to that which was expected in the case of aged persons (55–64 years, both total and female-only) by looking at their education level, the structure of the labour market, and the percentage of aged people paid with minimum wage. Another very interesting aspect could be the migration rate from the total active population.

By reconfirming the highly probable negative impact of minimum wage dynamics on the dynamics of employment, our research recommends more responsibility and a healthy dose of skepticism in the use of this public policy tool. In many countries, minimum wage is very politicized, populist, and arbitrarily used, without any impact assessment and without correlation with labour productivity or other labour market conditions. The minimum wage systems are today very differentiated at the level of the European countries, with so many features regarding the implication of market powers in this bargaining. The introduction of the EU minimum wage could further complicate this discussion, especially if the economic arguments are completely neglected.

**Author Contributions:** Conceptualization: all authors; Methodology: C.V.P.; Software (Eviews 12): C.V.P.; Validation: all authors; Formal analysis: all authors; Investigation: all authors; Resources: all authors; Data curation: C.V.P.; Writing—original draft preparation: C.V.P., A.P.; Writing—review and editing: all authors; Visualization: C.V.P.; Supervision: C.V.P.; Project administration: C.V.P. All authors have read and agreed to the published version of the manuscript.

**Funding:** This research received no external funding.

**Conflicts of Interest:** The authors declare no conflict of interest.

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
