# Peer review of "The Impact of the Minimum Wage on Employment: An EU Panel Data Analysis"

_sustainability, doi:10.3390/su13169359_

Round 1
Reviewer 1 Report
- The Literature review is very informative. However, it can be shortened and consolidated, especially towards the end. The paragraph about publication bias should be placed towards the end of the literature review as it is a nice wrap-up and gives a hint how to interpret the results presented in the literature. Spell-cecking is adviced!
- Section 2 starts with another literature review. This seems strange. Those citations should go to section 1.
- ad H3: But wages tend to increase with age (seniority). Is H3 a valid hypothesis?
- By assuming that young workers, old workers and women are hit more severely than others, you only assume that workers with lower wages might more easily be subject to the minimum wage. So, it all collapses to that single hypothesis.
- I do not get H5.
- You say, the countries in your sample are homogenous but then add a long list of differences with regard to minimum wage regulations.
- The data section is poorly formatted and the language becomes worse.
- Okay. I am not sure at all what you estimate and what models 1-8 refer to. Do I understand you correctly that the series are non-stationary and that you have cointegration (btw: How can employment numbers be non-stationary)? If so, adding lags to your variables won't solve the problem, right? An error correction model would be in order. And what is it with the FE/RE-section? What does this tell us? As I do not get your estimation approach (regardless of whether it's incorrect or poorly presented). The entire approach/results-section would need to be clarified.
Author Response
General answer of the authors to the Reviewer:
We are really appreciating the comments and the remarks. We find them very useful and constructive, contributing to the improvement of our paper. We highlighted with yellow color all the changes in this new version of the manuscript. This time, we formatted the text using the template of the journal.
REVIEWER: The Literature review is very informative. However, it can be shortened and consolidated, especially towards the end. The paragraph about publication bias should be placed towards the end of the literature review as it is a nice wrap-up and gives a hint on how to interpret the results presented in the literature. Spell-checking is advised!
The answer of the authors: We revised the literature review entirely. We shortened and consolidated it as much as possible. We moved the suggested paragraph with publication bias before the concluding remarks in the literature section, as you suggested. We recheck the whole text for English and spelling errors.
REVIEWER: Section 2 starts with another literature review. This seems strange. Those citations should go to section 1.
The answer of the authors: Initially we included this section to explain the research hypothesis better. However, to avoid confusion, we condensed it and moved it into the literature review section, as you suggested.
REVIEWR: ad H3: But wages tend to increase with age (seniority). Is H3 a valid hypothesis?
REVIEWER: By assuming that young workers, old workers, and women are hit more severely than others, you only assume that workers with lower wages might more easily be subject to the minimum wage. So, it all collapses to that single hypothesis.
The answer of the authors: We provided more explanations regarding hypothesis H3 in the new version of the manuscript. This dis-employment effect of the minimum wage for selected categories is present in the economic literature, empirical tests and can be theoretically argued for the following categories of employed persons: teens (after graduation they are costly for the company), women (pregnancy, limited physical capacity for some hard jobs, etc.), aged persons, low educated persons, a combination of all of these. This dis-employment effect does not refer only to the people fired by companies when this dynamic of the minimum wage cannot be supported by the dynamic of the value-added but also refers to the jobs that are not created by the private sector. The unsustainable dynamic of the minimum wage will determine companies to be reluctant to hire vulnerable persons too and this is captured by the dynamic of the employment. For this reason, the dynamic of employment is better than the unemployment rate. Moreover, sometimes, the dynamic of the minimum wage could also be a strong signal for the dynamic of all wages in the company/sector. As we found in other studies, we expected to obtain a negative impact on employment for this vulnerable category.
REVIEWER: I do not get H5
The answer of the authors: We detailed this hypothesis in the new version of the paper too. Normally, the minimum wage is a barrier for companies to employ labor from the market. If the dynamic is uncorrelated with the capacity of companies to pay this salary, the impact on the dynamic of the employment can last more than one year. The cointegration tests confirmed this long-term relationship and, for this reason, we decided to also test the impact of lagged minimum wage dynamic on the current employment’s dynamic.
REVIEWER: You say, the countries in your sample are homogenous but then add a long list of differences with regard to minimum wage regulations.
The answer of the authors: We eliminated that section. The systems are QUITE HOMOGENOUS. Of course, there are differences by countries that are also captured in our study by fixed effects that fitted the outputs of our panel regressions. To avoid any confusion, we eliminated this part.
REVIEWER: The data section is poorly formatted and the language becomes worse.
The answer of the authors: We significantly review this part. We included all necessary tables of outputs to support the conclusions and the findings. The data section and the results section were significantly reviewed as you suggested. Now, we consider that it is clearer for everybody what we did in this research.
REVIEWER: Okay. I am not sure at all what you estimate and what models 1-8 refer to. Do I understand you correctly that the series are non-stationary and that you have cointegration (btw: How can employment numbers be non-stationary)? If so, adding lags to your variables won't solve the problem, right? An error correction model would be in order. And what is it with the FE/RE-section? What does this tell us? As I do not get your estimation approach (regardless of whether it's incorrect or poorly presented). The entire approach/results section would need to be clarified.
The answer of the authors: In our study, we used log differences for all the variables included in the model (employment, minimum wage, labor productivity, and labor force). It can be associated with the first difference between these variables. The unit-root tests are now more clearly mentioned in the text. We used 8 data panels because we tested the impact of the dynamic of the minimum wage on the dynamic of employment for 8 different categories. The Hausman test indicated partially the presence of random effects but the Redundant Fixed Effects Tests clearly indicated the presence of fixed effects. Accordingly, with these results, we tested successively these effects to obtain the best-fitted estimators and, in the paper, we presented now only the final result. Most of them are only with Fixed effects only. However, we significantly changed this section, according to our suggestion, being now clearer for everybody. The results are consistent with previous studies and statistically significant.
We hope that we answered all your questions by the reviewed manuscript and we thank you for your very useful comments addressed to us.

Reviewer 2 Report
The paper "The impact of the minimum wage on employment: an EU panel data analysis" is interesting for journal readers but there are more points to be addressed.
The aim of the analysis should be evidenced in the abstract and introduction sections.
The interpretation of the results should be further discussed in terms of policy implications.
The conclusions should be improved with the weaknesses of the analysis and the insights for future research.
Author Response
REVIEWER: The paper "The impact of the minimum wage on employment: an EU panel data analysis" is interesting for journal readers but there are more points to be addressed. The aim of the analysis should be evidenced in the abstract and introduction sections.
The answer of the authors: We included in the abstract this suggestion.
- The interpretation of the results should be further discussed in terms of policy implications.
The answer of the authors: We included a final paragraph discussing the policy implications.
- The conclusions should be improved with the weaknesses of the analysis and the insights for future research.
The answer of the authors: In the conclusion section we marked the paragraph that discusses the limitations of the research and the further developments.
